# Skilled musicians are indeed subject to the McGurk effect

## Stephen Politzer-Ahles and Lei Pan

Department of Chinese and Bilingual Studies, The Hong Kong Polytechnic University, Hung Hom, Hong Kong

SP-A, 0000-0002-5474-7930

## Replication

psychology/cognition

McGurk effect, speech perception, audiovisual integration, musicians, replication

**Author for correspondence:**
Stephen Politzer-Ahles
e-mail: stephen.politzerahles@polyu.edu.hk

The McGurk effect is an illusion whereby speech sounds are often mis-categorized when the auditory cues in the stimulus conflict with the visual cues from the speaker's face. A recent study claims that 'skilled musicians are not subject to' this effect. It is not clear, however, if this is intended to mean that skilled musicians do not experience the McGurk effect at all, or if they just experience it to a lesser magnitude than non-musicians. The study also does not statistically demonstrate either of these conclusions, as it does report a numerical (albeit non-significant) McGurk effect for musicians and does not report a significant difference between musicians' and non-musicians' McGurk effect sizes. This article reports a pre-registered, higher-power replication of that study (using twice the sample size and changing from a between- to a within-participants manipulation). Contrary to the original study's conclusion, we find that musicians do show a large and statistically significant McGurk effect and that their effect is no smaller than that of non-musicians.

## 1. Introduction

When people comprehend physical stimuli, they integrate information from multiple sensory modalities to generate a psychological percept. For example, the way people recognize a sound can be modulated by visual sensory information accompanying the sound. Perhaps the most famous example of this is the McGurk effect [1], whereby people tend to mis-categorize speech sounds that are dubbed onto a video of people pronouncing a different sound. For example, people might accurately judge an audio recording of 'ba' as being the sound /ba/ (slashes indicate representations in the International Phonetic Alphabet), but when that same sound is dubbed over a video of a person saying 'ga' then people often judge the sound as being something other than 'ba'. This effect is strong evidence that information from one modality (visual) can influence the recognition of information in another modality (auditory).

Proverbio and colleagues [2] argue that the McGurk effect is absent in highly trained musicians. This observation is important because it suggests that their experience changes some

mechanisms of speech comprehension, such as, for example, the relative weighting of different types of cues (acoustic, phonetic, visual) when speech is difficult to perceive.

Some aspects of the study, however, limit the conclusiveness of this finding. While the experiment did not reveal a significant difference between musicians' accuracy in audiovisual 'McGurk' stimuli and audio-only stimuli, they nevertheless showed a large numerical effect in the direction of a typical McGurk effect (higher accuracy in audio-only than audiovisual incongruent stimuli). The experiment probably had low power to detect a significant effect, as this critical comparison was between only 10 participants who heard audio-only stimuli and 20 who saw audiovisual stimuli. Furthermore, the experiment also tested a control group of participants without musical experience, and there was not a significant interaction between the groups and the type of stimuli perceived. Without a significant interaction, the conclusion that non-musicians had a McGurk effect and musicians did not is not necessarily justified: the fact that one group shows a significant effect and another group does not show a significant effect is not, in of itself, sufficient evidence that the two groups are significantly different from one another [3]. Another limitation is that the experiment included many stimuli that would not necessarily be expected to elicit McGurk effects even in typical participants: for example, while an audio recording of 'ba' dubbed over a video of a person saying 'ga' presents the listener with visual cues that conflict with the auditory input, a recording of 'ba' dubbed over a video of a person saying 'pa' does not, as these two sounds share the same place of articulation and the main difference between them (the duration between the release of the consonant and the onset of vocal fold vibration) is not easily visible. Proverbio and colleagues [2] classified both types of stimuli as incongruent/McGurk audiovisual stimuli; the inclusion of stimuli like 'ba' audio dubbed over a 'pa' video may have caused McGurk effects to be underestimated. In fact, if their data are re-analysed with only cases of different place of articulation being included in the McGurk condition, the difference between musicians and non-musicians is reduced (musicians are 3.1% more accurate in the original analysis, but only 1.9% more accurate in this analysis). A final limitation is that the results reported in the paper do not match (i.e. cannot be replicated from) the data that are provided in the paper, as shown in comments on the online version of the paper.

Overall, the findings of Proverbio and colleagues [2] seem indeterminate: they are consistent with the presence of a McGurk effect for musicians and they are also consistent with the absence of such an effect. For these reasons, it is valuable to conduct a close replication of this study to assess whether skilled musicians really are not subject to the McGurk effect. For a stronger test of this question, we make several changes to the study design:

(1) We manipulate the crucial comparison (whether participants hear audiovisual stimuli or audio-only stimuli) within participants, rather than between participants, to have greater statistical power.
(2) We double the sample size. An exact power analysis was not possible, as the exact effect size and variance structure in Proverbio and colleagues' [2] data are unknown and their results not reproducible; thus, in the absence of knowing exactly how many participants are needed to detect a difference of a given size, we instead opted to increase power as much as possible. One way of increasing power is increasing the sample size, which other McGurk effect researchers have already noted is necessary [4].
(3) In an additional analysis, we quantify McGurk effects by classifying only stimuli in which the visual and audio place of articulation mismatch (e.g. a recording of 'ba' dubbed on a video of 'da') as McGurk stimuli, and classifying others as congruent audiovisual stimuli. This is another way to increase power, by improving the precision of the effect estimate (by not including observations which are not expected to show the effect). (Point #1 above is another way of increasing power by improving precision: using a within-participants manipulation helps reduce the influence of between-participants noise in the estimate.)
(4) We improve the statistical analysis by using generalized linear mixed models [5], rather than analysis of variance, which Proverbio and colleagues [2] used but which are inappropriate for binomial data of this sort [6]. This is yet another way to improve power, by using a more appropriate statistical model.
(5) We test two hypotheses of relevance. While Proverbio and colleagues [2] focus on whether musicians show a McGurk effect at all, the conceptual conclusions of their paper would also be supported if musicians just showed a smaller (but still significant) McGurk effect than non-musicians did. We thus report two comparisons: whether musicians' McGurk effect is greater than zero, and whether it is smaller than non-musicians'. Possible outcomes of the experiment are (a) musicians' McGurk effect is non-significant and smaller than that of non-musicians (consistent with the conclusions Proverbio and colleagues [2] made, although not fully consistent with their results); (b) musicians'

McGurk effect is significantly greater than zero but smaller than that of non-musicians (consistent with the conclusions and results of Proverbio and colleagues [2]); (c) musicians' McGurk effect is significantly greater than zero and not significantly smaller than that of non-musicians (inconsistent with the conclusions of Proverbio and colleagues [2], but consistent with their results); or (d) musicians' McGurk effect is not significantly different either from zero or from non-musicians.

# 2. Material and methods

All experimental methods were pre-registered at https://osf.io/cuzax/register/565fb3678c5e4a66b5582f67. In addition, the accepted Stage 1 manuscript (submitted after data collection via the Results Blind track) was separately registered at the point of Stage 1 in principle acceptance and is available at https://osf.io/xrphg/register/5a970dfec69830002df68ac2 .

## 2.1. Participants

Sixty-two skilled musicians and 62 non-musicians were recruited in Hong Kong and Shenzhen. (As Proverbio and colleagues [2] had 30 musicians and 30 non-musicians in the critical conditions, we set a goal of 60 musicians and 60 non-musicians; as we ran the participants in groups, we scheduled slightly more participants than necessary in case of no-shows.) Demographic details for the participants are available at https://osf.io/5ezcp/. Following Proverbio and colleagues [2], we recruited musicians who had at least 13 years' training in musical instruments and were not singers. The non-musicians were participants who reported that they did not listen to music for more than an hour per day and either had no music training at all or had not had music training within the past 10 years. All participants were native speakers of Mandarin or Cantonese.

While there have been reports that speakers of Chinese languages show less McGurk effect than English speakers [7], more recent studies have suggested that Chinese speakers show comparable McGurk effects to American English speakers [8,9]. If speakers of Chinese languages are less susceptible to McGurk effects (which does not seem likely), this would increase the chance of a floor effect (failing to see a difference between musicians and non-musicians because both groups' effects are so small). Thus, if anything, our sample biases our experiment in favour of supporting Proverbio and colleagues' [2] claim that musicians do not show McGurk effects.

## 2.2. Stimuli

All stimuli used in the experiment are available at https://osf.io/5ezcp/. Following Proverbio and colleagues [2], we chose eight consonants to use for the experiment: /b, p, m, f, t, d, l, k/. We limited ourselves to consonants that form existing morphemes in both Mandarin and Cantonese in the frame /_a/. We did not use /n/ because /n/ and /l/ are merged in many southern dialects of Mandarin and Cantonese. We also did not use /g/ because, while both /kaˈ/ and /gaˈ/ are not very meaningful in Mandarin (they are mainly used in phonetic borrowings, like 咖啡 /kaˈ feiˈ/ 'coffee'), /kaˈ/ is much more frequent.

We recorded video and audio of each sound being produced by one female native speaker of Mandarin (the second author of this paper, who also served as the experimenter collecting the data) in front of a dark blue background. Each sound was produced three times with an approximately 1 s stimulus onset asynchrony (realized by showing the speaker prompts with that timing), and several seconds of silence before and after; four such trios of each sound were recorded, and the best trio (with the clearest productions and least background noise) for each sound was selected. The videos with sound were segmented to have 1 s of silence before and after each trio. Then, incongruent McGurk audiovisual stimuli were created by replacing each video's soundtrack with each other sound's soundtrack (we used Praat [10] to adjust the timing of sounds such that the onset of each sound aligned with the onset of articulation in the video). Microsoft Movie Maker was used to replace soundtracks. This procedure yielded 64 videos (eight sounds times eight videos), of which eight were completely congruent (original, un-edited videos) and 56 were edited to replace the original sound with a new sound. Finally, the videos were edited to include a fixation cross located approximately at the tip of the speaker's nose.

The same procedure was used to create practice stimuli, using the consonants /w/ and /j/.

## 2.3. Procedure

Stimulus presentation and response logging were controlled using DMDX [11] (stimuli and scripts available at https://osf.io/5ezcp/). The 64 audiovisual stimuli and 64 audio-only stimuli were arranged in random orders (62 different stimulus lists with their own randomized orders were created, and each list was used for one musician and one non-musician); after each stimulus, the participant was prompted to use the keyboard to enter their transcription of what sound they believe they heard. Participants were instructed to focus on the fixation cross on the screen. Prior to the main experiment, the four audiovisual practice trials and four auditory-only practice trials were presented in a fully random order.

## 2.4. Analysis

Data were automatically coded as correct or incorrect based on whether the first character of the response (forced to uppercase, and with leading whitespaces trimmed) matched the first character of the expected correct response (e.g. for a stimulus whose auditory sound we coded as 'KA', responses of 'KA', 'ka', 'KAA' or 'KO' would all be marked as correct, but 'GA' or 'TA' would not). (This deviates from the pre-registered plan, where we stated that the responses would be manually coded as 'correct' or 'incorrect' by the authors. We opted for automatic coding instead because it is far more efficient and, based on cursory reviewing of several responses, accurate.)

Statistical analysis was conducted using generalized (logistic) linear mixed-effects models [4] with random effects for participants; maximal random slopes justified by the design were used [12]. The models were fitted using the lme4 package [13] of the R statistical computing environment [14]; all analysis code is available at available at https://osf.io/5ezcp/. We conduct two analyses: a replication analysis meant to closely replicate the comparisons made by Proverbio and colleagues [2], and a targeted analysis focusing on comparisons where there was a stronger *a priori* expectation of observing McGurk effects (as described in point (3) above). The details of the implementation of each analysis are described in the Results.

# 3. Results

All data files are available at https://osf.io/5ezcp/.

For each analysis, we used mixed-effects models to compare the likelihood of correct responses across conditions and groups. The difference between analyses lies in what trials are assigned to which condition: for the replication analysis, trials with the same place of articulation for the audio and visual stimuli but different voicing or manner (e.g. audio 'ba' with visual 'pa') were treated as incongruent trials, whereas for the targeted analysis, these were treated as congruent trials and thus were not analysed (see below).

Each analysis compared accuracy in the audio-only and audiovisual incongruent conditions, ignoring the audiovisual congruent condition. (Proverbio and colleagues' [2] key analysis comparing musicians and non-musicians is based on comparing audio-only and audiovisual incongruent conditions, as shown in their figure 1 and the first paragraph of their Results section. The only time they use the audiovisual congruent condition to quantify McGurk effects is in follow-up analyses examining interactions with different phonemes (their figures 2 and 3).) The McGurk effect was quantified as the difference in accuracy between the audiovisual incongruent condition and the audio-only condition. In each analysis, we regressed accuracy on condition (audio-only versus audiovisual incongruent), group (musicians versus non-musicians) and their interaction, as well as random intercepts for participants and random by-participant effects of condition (including correlations between these slopes and intercepts); full analysis code and model specifications are available at https://osf.io/5ezcp/. For each analysis, we focus on two comparisons, corresponding to the two hypotheses listed in point (5) in the Introduction: the McGurk effect for musicians (simple effect comparing accuracy for musicians in the audio-only condition to accuracy for musicians in the audiovisual incongruent condition), and the difference between musicians' and non-musicians' McGurk effects (the interaction coefficient). Group was dummy-coded with musicians as the baseline level, and condition dummy-coded with audio-only as the baseline level. Therefore, the coefficient for the 'condition' factor represents the McGurk effect for musicians and will be negative if audiovisual incongruent trials have lower accuracy than audio-only trials; and the interaction coefficient will be negative if musicians have a smaller McGurk effect than non-musicians.

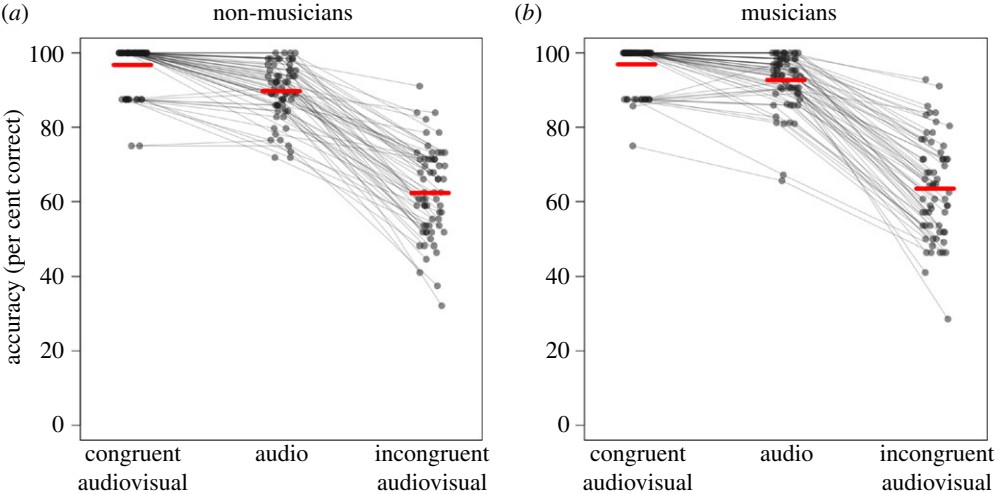

**Figure 1.** Results for each participant, with conditions coded according to Proverbio and colleagues' [2] scheme. (*a*) Shows the results for non-musicians, and (*b*) results for musicians. In each panel, accuracy for congruent audiovisual stimuli is shown on the left, followed by accuracy for auditory-only stimuli in the middle and accuracy for incongruent audiovisual stimuli on the right. The *y*-axis shows accuracy ranging from 0% correct at the bottom to 100% correct at the top. In each panel, each participant's accuracy on the three conditions is represented by a series of three dots connected by lines. Superimposed over the dots in each condition is a thick red line, one per condition, showing average accuracy across participants. For each group, accuracy in the congruent audiovisual condition is clustered near the maximum, accuracy in the audio-only condition is similar or slightly lower for most participants and accuracy in the audiovisual incongruent condition is slightly or substantially lower than accuracy in the audio-only condition for every participant.

## 3.1. Replication analysis

A summary of the results is shown in figure 1. It is clear that both musicians and non-musicians had robust McGurk effects: for non-musicians, the audiovisual incongruent condition is 27.4 percentage points less accurate than the audio-only condition, and for musicians, it is 29.2 percentage points less accurate (95% two-tailed percentile bootstrap confidence interval: 27.7%–33.4%). The McGurk effect for musicians is numerically *larger* than that for non-musicians (95% CI of difference: −2.5%–5.7%). The statistical analysis confirmed that musicians have a highly significant McGurk effect ($b = -2.34$, $z = -18.37$, $p < 0.001$), based on the simple effect coefficient for stimulus condition (which refers to the difference between audio-only and audiovisual incongruent for musicians, since musicians were coded as the baseline level of the group factor). Furthermore, musicians' McGurk effect was in fact significantly larger than the non-musicians' McGurk effect ($b = 0.46$, $z = 2.72$, $p = 0.007$), as shown by the condition–group interaction coefficient. (While the significant model coefficient for this effect may appear to contradict the confidence interval reported above, which includes 0%, the source of this apparent discrepancy is that the reported confidence interval is converted into percentage units, whereas the model coefficient and associated *p*-value is based on odds ratios, which are more sensitive to differences in probability. The calculation of these values is shown in the data analysis code on OSF.)

## 3.2. Targeted analysis

A summary of the results is shown in figure 2. The overall pattern is similar to that of the replication analysis (figure 1), except that accuracy on the incongruent audiovisual conditions is visibly lower since this condition no longer includes stimuli with the consistent place of articulation between video and audio. Non-musicians have a McGurk effect of 37.3 percentage points, and musicians 38.3 percentage points; again the effect for musicians is numerically larger than that for non-musicians. The statistical analysis confirmed that musicians have a highly significant McGurk effect (simple effect coefficient of stimulus condition: $b = -2.73$, $z = -20.89$, $p < 0.001$) and that their McGurk effect is significantly higher than that for non-musicians (interaction coefficient: $b = 0.42$, $z = 2.44$, $p = 0.015$).

As an exploratory analysis, we also indirectly compared the two coding schemes for audiovisual stimuli. As mentioned above, the targeted analysis (treating stimuli like audio 'ba' with visual 'pa' as congruent) yielded numerically larger McGurk effects than the original study's analysis (treating such

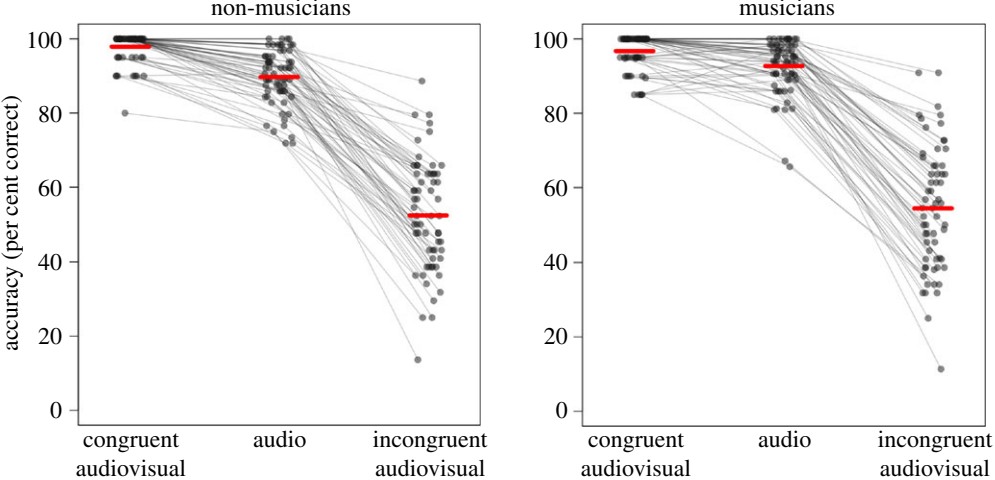

**Figure 2.** Results for each participant, with conditions coded in our targeted analysis (based only on whether the place of articulation is incongruent between audio and visual information). The figure is laid out in the same way as figure 1, and the pattern of results is the same, except that the accuracy for congruent audiovisual trials is generally higher and the accuracy for incongruent audiovisual trials generally lower.

stimuli as incongruent). While comparing these two analysis techniques was not part of our pre-registered plan (nor part of the approved analysis plan for this registered report) and thus strong conclusions should not be made from statistical significance patterns in this analysis, it nevertheless may be useful as a source of further information about how McGurk effect coding should be done in the future. Thus, we compared the accuracy on ambiguous stimuli (those that we treated as congruent while Proverbio and colleagues [2] treated as incongruent) to the accuracy for unambiguously incongruent and unambiguously congruent stimuli. Based on our intuition, the reason that the McGurk effect is bigger in the targeted analysis than in the replication analysis should be that the replication analysis treated those ambiguous stimuli as incongruent, whereas they actually behave like congruent stimuli (and thus in the replication analysis they artificially brought up the accuracy for the incongruent condition, whereas in the targeted analysis they do not). Thus, we expect that the ambiguous stimuli should be responded to more accurately than incongruent stimuli and not less accurately than congruent stimuli. In a mixed-effects model (with stimulus type dummy-coded with ambiguous stimuli as the baseline, so it could be directly compared to the other two categories), this pattern was indeed observed: the incongruent stimuli were responded to significantly less accurately than the ambiguous stimuli ($b = -4.36$, $z = -10.87$, $p < 0.001$), whereas the congruent stimuli did not significantly differ from the ambiguous stimuli ($b = -0.57$, $z = -1.16$, $p = 0.244$).

## 4. Discussion

In a pre-registered study with more than 62 musicians and 62 non-musicians (compared to 30 musicians and 30 non-musicians in the original study), we found that musicians are subject to the McGurk effect, and that their susceptibility to the McGurk effect is not significantly less than that of non-musicians—if anything, it is greater. These results are in fact broadly consistent with the results, but not the conclusions, reported by Proverbio and colleagues [2]. They also found that musicians were numerically more accurate on audio-only than incongruent stimuli, and although this effect was not significant, the low sample size and wide error bars in their figure 1 suggest that this effect probably has a wide confidence interval, making it also not inconsistent with a McGurk effect. Likewise, they did not find a significant interaction between participant group and stimulus condition, but the likely wide confidence interval of this effect in their results makes it not inconsistent with our results, in which musicians' McGurk effects were numerically quite close to (albeit statistically significantly larger than) non-musicians'. Thus, both their results and ours are consistent with the conclusion that musicians are subject to the McGurk effect at least as much as non-musicians are; our study provides additional evidence for this conclusion with higher power and more precise statistical estimates of the effect. Figure 3 shows a schematic comparison of our results and those of Proverbio and colleagues [2] (note that this figure is not showing exact values, but an idealized schematic).

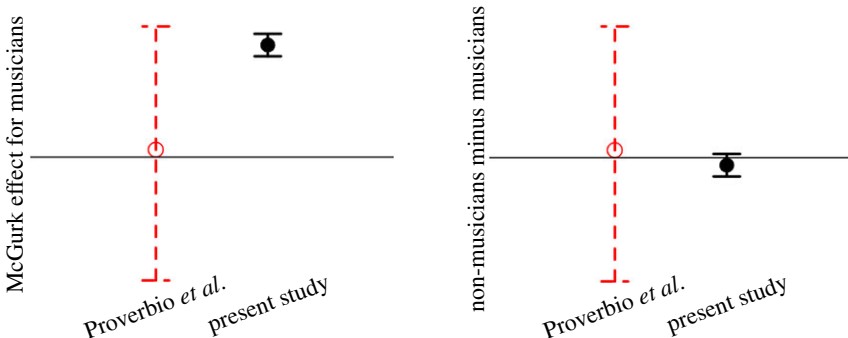

**Figure 3.** Schematic comparison of the results of this study with the results of Proverbio *et al.* [2]. In each plot, a rough guess of the effect and confidence interval from Proverbio *et al.* [2] is shown; exact effects cannot be plotted because the exact values of the effects in the paper are not clear (see text). Likewise, the effect and confidence interval of this study are plotted. The left-hand plot shows the McGurk effect for musicians (represented as accuracy in the audio-only condition minus accuracy in the audiovisual incongruent condition, such that higher values represent larger McGurk effects; the horizontal line represents zero McGurk effect). The effect observed by Proverbio and colleagues [2] was small and positive, but not significant, and probably had a large confidence interval (meaning it is not inconsistent with a large McGurk effect or with no McGurk effect). The effect observed in this study was large and positive, with a small confidence interval. It is not known whether the present effect is or is not within the confidence interval of Proverbio *et al.* [2] (see text), but we drew the schematic this way to represent that our result is not conceptually inconsistent with theirs. The right-hand plot shows the difference between non-musicians' and musicians' McGurk effects, such that a higher value represents bigger McGurk effects for non-musicians. The difference observed by Proverbio and colleagues [2] was small and positive, but not significant, and probably had a large confidence interval. The difference in this study was small and in the opposite direction, with a small confidence interval. Again, it is not known whether our effect is within the confidence interval of Proverbio *et al.* [2].

For comparing replication studies to original studies, comparing the pattern of significance is not the best approach [15]; it is better to perform some kind of direct statistical comparison between the two studies (ideally, between the effect observed in one study and a hypothetical effect size that the other study would have had sufficient power to detect). Simonsohn [15] outlines several ways this can be done, as well as their limitations. In the present case, how a direct comparison between studies would be conducted is not straightforward, because of differences between the analyses and because of the inability to replicate the analysis of the original study.

Our statistical results are based on mixed-effect models and Proverbio and colleagues' [2] on analysis of variance using arcsin-transformed data. Thus, to perform a direct statistical comparison, one of these datasets would need to be re-analysed. We were unable to re-analyse Proverbio and colleagues' [2] data using mixed-effects models: while they do provide fairly detailed tables summarizing the data (Proverbio *et al.* [2], tables 1 and 2), these tables do not indicate which responses come from which participants, and without that information, it is not possible to model these data with mixed effects. Furthermore, their tables 1 and 2 only show data for audiovisual conditions (congruent and incongruent), not for audio-only conditions.

The other option would be to analyse our data using the approach used by Proverbio and colleagues [2]; even though a mixed-effects model approach is more appropriate for these data (as explained in the Introduction), re-analysing our data using their approach would allow us to perform a direct statistical comparison. This was not possible, however, as their analysis is not replicable. As detailed in online comments on Proverbio *et al.* [2], there are discrepancies between the values reported in the article and the values calculated from the data provided according to the methods described in the article (i.e. the values in their data tables 1 and 2 do not add up to the values reported in their prose results or key figures—for instance, Proverbio and colleagues [2] report 93.75% accuracy for musicians in the congruent audiovisual stimuli whereas the congruent audiovisual cells [the diagonal] in table 1 average out to 96.875% accuracy). Therefore, it is not clear which of their reported values our results should be compared to.

Without being able to perform a direct statistical comparison between the studies, we just make rough comparisons between the raw values reported in the different studies, without making any claims about statistically significant differences between studies. According to the prose results and tables 1 and 2 in Proverbio *et al.* [2], musicians were 90% accurate on audio-only stimuli and 92% accurate on incongruent audiovisual stimuli, yielding a 2% effect in the opposite direction of a typical McGurk effect. On the other hand, their figure 1, showing arcsin-transformed data, shows an effect numerically consistent with a

**Table 1.** Summary of differences between Proverbio *et al.* [2] and this study. 'Minor differences' are those which we have no reason to believe would substantially affect the results, but which we nonetheless report here in an attempt to be comprehensive, and with no intention to claim that one approach or the other is superior. Here we have not listed extremely minor procedural differences (such as the type of hardware used or the use of PowerPoint versus DMDX for stimulus presentation) which we have no *a priori* reason to believe would affect the results at all.

|  | Proverbio *et al.* [2] | this study |
| --- | --- | --- |
| major differences | | |
| pre-registered? | No | Yes |
| sample size | 30 musicians and 30 non-musicians | 62 musicians and 62 non-musicians |
| design | between participants (for audio-only versus audiovisual comparison) | within participants (for audio-only versus audiovisual comparison) |
| key comparisons | some conclusions based on comparing incongruent condition to audio-only, some on comparing incongruent to congruent | all comparisons based on comparing incongruent to audio-only |
| stimuli | little background noise | substantial background noise |
| condition coding | stimuli with same place of articulation can be coded as incongruent | stimuli with same place of articulation coded as congruent (in the second analysis only) |
| analysis | ANOVA on arcsin-transformed data | generalized linear mixed-effects models |
| data availability | summary data in article | all data and code in online repository |
| McGurk effect for musicians? | unclear (but reported as no effect) | large effect |
| musicians' McGurk effect smaller than non-musicians'? | unclear (effect numerically smaller but not significant) | no (actually significantly *larger*) |
| minor differences | | |
| non-musicians' inclusion criteria | 'lack of musical studies and specific interest in music as a hobby' | no music training within the past 10 years |
| language | Italian | Mandarin and Cantonese |
| inter-trial interval | 5 s | 4 s |
| participant distance from monitor | 80 cm | not controlled |
| individual or group administration | not reported | participants came in groups |
| response coding | manual | automated |

McGurk effect (musicians about 2.5 percentage points more accurate with audio-only than audiovisual incongruent stimuli); it is not clear how arcsin transformation would reverse the sign of the effect. It is also possible that one or more of these values may be inaccurate, given other inconsistencies in the data reporting mentioned above. Non-musicians, on the other hand, were 94% accurate on audio-only stimuli and 89% accurate on incongruent audiovisual stimuli, leading to a McGurk effect of 5% (the same caveat about inconsistencies in the reported values for congruent audiovisual stimuli also applies here). Keeping in mind the caveats associated with these reported values, we roughly compare those McGurk effect sizes to the ones obtained in this study.

Proverbio and colleagues' [2] reported McGurk effects of −2% (or+2.5%) and 5% are far smaller than the McGurk effects of 27% and 29% observed in our study (see 'Replication analysis' above), which may

in part be due to our use of noisier recordings, moving the participants farther away from ceiling. If these results are taken at face value, it seems that our results are inconsistent with theirs: the McGurk effect observed in musicians is far smaller in their study than ours, and the difference between musicians' and non-musicians' McGurk effects in their study is opposite the corresponding difference in ours. The difference between musicians' and non-musicians' McGurk effects in their study, however, might be just barely outside the confidence interval of the corresponding difference in our study (depending on what the McGurk effect size for musicians in their study was), and the difference in our study is probably within the confidence interval of theirs (given the smaller sample size and between-participants comparison of their study, their confidence interval is probably wider than ours). Furthermore, given that our study is higher powered (based on the larger sample size and the use of within-subject comparisons) and pre-registered, and given the other improvements made in our study (listed in the Introduction), we are confident that our result—musicians showing a substantial McGurk effect, which is not significantly smaller than that shown by non-musicians—the more robust one.

One unplanned methodological difference between this study and that of Proverbio and colleagues [2] is the presence of background noise in the stimuli: our recordings were made in a noisier environment than theirs. Nonetheless, we do not believe this difference can account for the difference in the two studies' results. First of all, the noise is the same in all conditions, so if it worsens perception in one condition, we would expect it to also do so in the others; thus, it presumably would not have created a spurious McGurk effect or eliminated a real one (although it may amplify the size of the effect, as mentioned above, by moving the participants' accuracy further away from ceiling, leaving more room to detect a McGurk effect which may otherwise have been obscured by a ceiling effect). Most importantly, the noise is also the same for musicians and non-musicians and thus does not confound the comparison between these groups. If anything, if musicians are better than non-musicians at processing speech in noise [16], then they might show less reliance on supporting visual cues and thus less McGurk effect than non-musicians (which would be consistent with the observation of Hirst and colleagues [17] that susceptibility to the McGurk effect is modulated by the extent to which the perceiver is dominant in auditory versus visual perception; we thank an anonymous reviewer for bringing this argument to our attention). In our study, the difference between musicians' and non-musicians' McGurk effects was in the opposite direction from this (musicians showed more McGurk effect, not less), and thus we do not see any way that the background noise in the stimuli could account for this. In any case, even if the difference in background noise could explain the difference between the two studies' results in some way we have not yet considered, this would suggest that the conclusions of Proverbio and colleagues [2] are contingent on a hidden moderator such as noise level (although this speculation would need to be tested in confirmatory research comparing the musician versus non-musician difference with and without background noise in the same study), which would also be important information for the field to know.

The differences between this study and that by Proverbio and colleagues [2] are summarized in table 1.

Proverbio and colleagues [2] make several conclusions about how musicians' non-susceptibility to the McGurk effect suggests that their music experience may have re-organized the functional specification of several brain areas. The present study obviously rules out those conclusions: if musicians *are* indeed subject to the McGurk effect, then McGurk studies do not provide any evidence for that type of brain-level functional reorganization as a result of musical training. If there is functional cortical reorganization in musicians, it may be of a different nature than that suggested by Proverbio and colleagues [2]: as suggested by the present results, the difference between musicians and non-musicians (if any) is that musicians are, if anything, *more* susceptible to the McGurk effect, not less. (This difference should be interpreted with caution, however, as it was not predicted and thus needs further replication.) The present results do not necessarily, however, challenge the overall notion of brain plasticity as a result of training or experience; as reviewed by Proverbio and colleagues [2] in their discussion, there are several other lines of research providing converging evidence for the notion of plasticity in musicians, and thus this study does not alone reject that entire notion. It does suggest, however, that McGurk effects are not a piece of evidence for that sort of plasticity.

# 5. Conclusion

In short, the most parsimonious conclusion of the results from Proverbio *et al.* [2] and this study is that musicians *are* subject to the McGurk effect, and to at least as much extent (if not more) as non-musicians are.

Ethics. Experimental procedures were approved by the Human Subjects Ethics Sub-Committee at the Hong Kong Polytechnic University (protocol number HSEARS20170806001) and the experiment was performed in accordance

with the Declaration of Helsinki. Participants provided informed consent and were reimbursed with cash for their participation.

Data accessibility. All stimuli, experiment programs, data and analysis code are available at https://osf.io/5ezcp/.

Authors' contributions. S.P.-A. and L.P. conceptualized the experiment; L.P. prepared the stimuli; S.P.-A. and L.P. prepared the experiment protocol; L.P. collected the data; S.P.-A. analysed the data; S.P.-A. and L.P. wrote the manuscript. Both authors gave final approval for publication.

Competing interests. The authors declare no competing interests.

Funding. L.P. was supported by grant #1-ZE89 from the Hong Kong Polytechnic University Faculty of Humanities Dean's Reserve. This paper also received support from the HK PolyU-PKU Research Centre on Chinese Linguistics.

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
