## [Reviewer comments · Royal Society Open Science]

Review History

RSOS-181868.R0 (Original submission)

Review form: Reviewer 1 (Rebecca Hirst)

Do you have any ethical concerns with this paper?

No

Have you any concerns about statistical analyses in this paper?

No

Recommendation?

Accept with minor revision

Comments to the Author(s)

Reviewer: Rebecca Hirst (University of Nottingham, UK)

Thank you for allowing me the opportunity to review this paper. The manuscript is very clearly written and easy to follow. It is clear how this study builds upon existing literature to answer an interesting question. The pre-registration and organisation of supplementary material and data

are also very clearly presented for future researchers to use. I only have two minor comments/suggestions:

1. For the analyses in this paper the McGurk effect is quantified as the difference between incongruent and auditory only. However, it looks as though Proverbio et al also compared congruent and incongruent trials (in addition to incongruent vs auditory only). A short justification as to why this effect was not explored in this study could make it easier to compare between studies.

2. I was unable to open the .zil data files on my mac operating system. Perhaps providing data in a second format (or providing info on how to open the .zil files - even if it means the user should find a windows computer) would increase data accessibility further (although another format may not work with the available R scripts?).

Nevertheless, from what I have read, I would recommend this paper for publication, and I look forward to reading the results.

Review form: Reviewer 2 (Alice Mado Proverbio)

Do you have any ethical concerns with this paper?

No

Have you any concerns about statistical analyses in this paper?

No

Recommendation?

Reject

Comments to the Author(s)

Revision of paper: Are skilled musicians subject to the McGurk effect?

The paper presented is full of falsehood, imprecisions, misunderstandings and incorrect statements referred to Proverbio et al. (2016) study. Plus, there is a complete lack of theoretical knowledge. Sounds and phonemes are not the same thing!! Also stimuli used in the two studies were different, and, especially, Proverbio et al's stimuli were not noisy.

Abstract

The authors stat that "sounds are often mis-perceived" in McGurk effect. This is incorrect.

Sounds are perceived fine, phonemes are mis-catagorized

The authors state in the abstract:

"it is not clear, however, if this is intended to mean that skilled musicians do not experience the McGurk effect at all, or if they just experience it to a lesser magnitude than non-musicians"

This is not scientific language!! The paper by Proverbio et al. (2016) showed that:

<<The analyses of data performed as a function of the phonemes auditorily perceived showed the statistical significance of "condition" factor (congruent vs. incongruent conditions) ($F_{8,112} = 3.646$; $p < 0.0008$), but only for controls. The analysis of group effects and Tukey post-hoc tests showed no significant difference between the congruent and the McGurk conditions in musicians but only in controls.>>

In scientific terms: there was no difference in the ability to correctly categorize phonemes in the AV congruent vs incongruent condition (meaning at all). The tendencies visible in the figure but not statistically significant cannot be considered as "to a lesser magnitude", this is not the scientific methods

The authors state: "The study also does not statistically demonstrate either of these conclusions" This is totally false, as explained below, and should be imperatively changed!

The authors state: "as it does report a numerical (albeit non-significant) McGurk effect for musicians, and does not report a significant difference between musicians' and non-musicians' McGurk effect sizes."

Again this is not correct, and, indeed, false. Proverbio et al. (2016) showed a significant decrease in the ability to correctly categorize phonemes in controls but NOT in musicians, and reported all the statistics!

<<as a function of the phonemes auditorily perceived showed the statistical significance of "condition" factor (congruent vs. incongruent conditions) ($F_{8,112} = 3.646$; $p < 0.0008$), but only for controls
s a function of the labial (lip movements) perceived was statistically significant ($F_{8,112} = 2.685$; $p < 0.0097$), but only for controls.>>

ABSTRACT. The authors state "and changing from a between- to a within-participants manipulation).!!

In Proverbio et al et al. 2016 the manipulation was WITHIN participants. ???

Introduction

The authors state: "People comprehend physical stimuli, they integrate information from multiple sensory modalities to generate a psychological percept. "

This is too generic, include appropriate scientific quotations relative to AV integration processes. Again please use a scientific language!

For example, the way people perceive a sound can be modulated by visual sensory information accompanying the sound. Perhaps the most famous example of this is the McGurk effect, whereby people tend to mis-perceive sounds.

This is incorrect. The McGurk effect has nothing to do with "sounds", but with phoneme categorizations. The two processes are completely different (please read and include some literature)

Line 42 "experience changes some mechanisms of basic sound perception" again there are many evidences that musical ability modify linguistic perception and ability. We are not talking only about a finer acoustic ability

Page 2 (lines 47-51) the experiment did not reveal a significant difference between musicians' accuracy in audiovisual "McGurk" stimuli and audio-only stimuli, they nevertheless showed a large numerical effect in the direction of a typical McGurk effect (higher accuracy in audio-only than audiovisual stimuli).

This is profoundly incorrect and false, and should be imperatively changed!

In Proverbio et al. (2016) study, both musicians and controls showed no difference between the auditory unimodal and audiovisual congruent conditions
(Musicians: visual = 10%; auditory = 90%; audiovisual = 93.75%. Controls: visual = 11.25%; auditory = 93.75%; audiovisual = 95%)

The authors state on page 2 (lines 51-52) the experiment likely had low power to detect a significant effect, as this critical comparison was between only 10 participants who heard audio-only stimuli and 20 who saw audiovisual stimuli

That's' totally incorrect and false. In Proverbio et al. (2016) say (right before the results) "In this study, 80 healthy, age-matched, graduate male and female volunteers were tested both in multimodal (McGurk condition) and unimodal (only auditory and only visual) conditions."

Did you even read the paper?!

The authors state on line 27 page 4: "As Proverbio and colleagues had 30 musicians and 30 non-musicians"

This is again incorrect!

The stimuli used were different from Proverbio et al (2016)

The authors state: "we limited ourselves to consonants that form existing morphemes in both Mandarin and Cantonese in the frame /_a/. We did not use /n/ because /n/ and /l/ are merged in many southern dialects of Mandarin and Cantonese. We also did not use /g/ because, while both /kaɿ / and /gaɿ / are not very meaningful in Mandarin (they are mainly used in phonetic borrowings, like 咖啡/kaɿ fei ɿ / coffee"), /ka ɿ / is much more frequent."

The stimuli used by Stephen Politzer-Ahles & Lei Pan are full of acoustic noise, and this is absolutely not comparable to the experimental conditions used by Proverbio et al (2016) study in which there was silence in background! Phoneme perception is worsened by the noise presence.

Review form: Reviewer 3

Do you have any ethical concerns with this paper?

No

Have you any concerns about statistical analyses in this paper?

No

Recommendation?

Accept in principle

Comments to the Author(s)

The authors have presented a plan to replicate a recent paper that compared the McGurk effect between musicians and non-musicians in Italy. The methods are straightforward and I think the original manuscript is a good one to attempt to replicate.

I am overall in favor of this replication attempt, but think the authors should do a bit more work to emphasize the generalizability of the current findings. Specifically, a power analysis of sorts would help determine if even n=60 per group is adequate to find small effects. As the authors' are aware from the Gelman articles they cite, a problem with studying small effects with small sample sizes is that any statistically significant effect tends to be exaggerated -- see also [1] below for application of this idea to between-group studies of the McGurk effect. This is important to ensure that these replication results are more decisive.

Second, the authors note that the "ga" syllable was not used (or will not be used), because of the subjects' native language. References concerning the robustness of the McGurk effect in their chosen population should be included to address head on any possibility of floor effect hiding any potential group effect.

[1] Magnotti JF, Beauchamp MS (2018) Published estimates of group differences in multisensory integration are inflated. PLoS ONE 13(9): e0202908.
<https://doi.org/10.1371/journal.pone.0202908>

[2] Sekiyama, K (1997). Cultural and linguistic factors in audiovisual speech processing: The McGurk effect in Chinese subjects. *Perception & Psychophysics*, 59(1), 73-80.

[3] Chen TH, & Massaro DW (2004). Mandarin speech perception by ear and eye follows a universal principle. *Perception & psychophysics*, 66(5), 820-836.

[4] Magnotti JF, Mallick DB, Feng G, Zhou B, Zhou W, & Beauchamp, MS (2015). Similar frequency of the McGurk effect in large samples of native Mandarin Chinese and American English speakers. *Experimental brain research*, 233(9), 2581-2586.

Decision letter (RSOS-181868.R0)

03-Dec-2018

Dear Dr Politzer-Ahles,

The Editors assigned to your Stage 1 Replication submission ("Are skilled musicians not subject to the McGurk effect?") have now received comments from reviewers. We would like you to revise your paper in accordance with the referee and editors suggestions which can be found below (not including confidential reports to the Editor). Please note this decision does not guarantee eventual acceptance.

Please submit a copy of your revised paper within three weeks (i.e. by the The author due date is unavailable). If deemed necessary by the Editors, your manuscript will be sent back to one or more of the original reviewers for assessment. If the original reviewers are not available we may invite new reviewers.

When submitting your revised manuscript, you must respond to the comments made by the referees and upload a file "Response to Referees" in the "File Upload" step. Please use this to document how you have responded to the comments, and the adjustments you have made. In order to expedite the processing of the revised manuscript, please be as specific as possible in your response.

Once again, thank you for submitting your manuscript to Royal Society Open Science and I look forward to receiving your revision. If you have any questions at all, please do not hesitate to get in touch. Full author guidelines may be found at <http://rsos.royalsocietypublishing.org/page/replication-studies#AuthorsGuidance>.

Kind regards,
Professor Chris Chambers
Royal Society Open Science
openscience@royalsociety.org

on behalf of Professor Chris Chambers (Registered Reports Editor, Royal Society Open Science)
openscience@royalsociety.org

Associate Editor Comments to Author (Professor Chris Chambers):

Associate Editor: 1

Comments to the Author:

Three expert reviewers have now assessed your submission. Reviewers 1 and 3 are positive and recommend in principle acceptance (IPA) after addressing some discrepancies between the analyses in the replication study compared with the original study, the rationale for the sample size (and consequent statistical power), concerns about the eliciting stimuli, and the accessibility of key materials. Reviewer 2, however, is much more critical, recommending rejection. The reviewer points to key differences between the replication study compared with the original study, as well as apparent errors of fact in describing the original study. Given the positive nature of reviews 1 and 3, I am going to invite a major revision. Please note that to achieve Stage 1 IPA, it is imperative that these issues are either resolved or sufficiently rebutted. Concerns about major deviations from the original methodology are the most serious issue, as the Replications format at RSOS is designed for close replications rather than conceptual replications.

Comments to Author:

Reviewer: 1

Comments to the Author(s)

Reviewer: Rebecca Hirst (University of Nottingham, UK)

Thank you for allowing me the opportunity to review this paper. The manuscript is very clearly written and easy to follow. It is clear how this study builds upon existing literature to answer an interesting question. The pre-registration and organisation of supplementary material and data are also very clearly presented for future researchers to use. I only have two minor comments/suggestions:

1. For the analyses in this paper the McGurk effect is quantified as the difference between incongruent and auditory only. However, it looks as though Proverbio et al also compared congruent and incongruent trials (in addition to incongruent vs auditory only). A short justification as to why this effect was not explored in this study could make it easier to compare between studies.

2. I was unable to open the .zil data files on my mac operating system. Perhaps providing data in a second format (or providing info on how to open the .zil files - even if it means the user should find a windows computer) would increase data accessibility further (although another format may not work with the available R scripts?).

Nevertheless, from what I have read, I would recommend this paper for publication, and I look forward to reading the results.

Reviewer: 2

Comments to the Author(s)

Revision of paper: Are skilled musicians subject to the McGurk effect?

The paper presented is full of falsehood, imprecisions, misunderstandings and incorrect statements referred to Proverbio et al. (2016) study. Plus, there is a complete lack of theoretical knowledge. Sounds and phonemes are not the same thing!! Also stimuli used in the two studies were different, and, especially, Proverbio et al's stimuli were not noisy.

Abstract

The authors stat that "sounds are often mis-perceived" in McGurk effect. This is incorrect. Sounds are perceived fine, phonemes are mis-catagorized

The authors state in the abstract:

“it is not clear, however, if this is intended to mean that skilled musicians do not experience the McGurk effect at all, or if they just experience it to a lesser magnitude than non-musicians”

This is not scientific language!! The paper by Proverbio et al. (2016) showed that:

<<The analyses of data performed as a function of the phonemes auditorily perceived showed the statistical significance of “condition” factor (congruent vs. incongruent conditions) ($F_{8,112} = 3.646$; $p < 0.0008$), but only for controls. The analysis of group effects and Tukey post-hoc tests showed no significant difference between the congruent and the McGurk conditions in musicians but only in controls.>>

In scientific terms: there was no difference in the ability to correctly categorize phonemes in the AV congruent vs incongruent condition (meaning at all). The tendencies visible in the figure but not statistically significant cannot be considered as “to a lesser magnitude”, this is not the scientific methods

The authors state: “The study also does not statistically demonstrate either of these conclusions”

This is totally false, as explained below, and should be imperatively changed!

The authors state: “as it does report a numerical (albeit non-significant) McGurk effect for musicians, and does not report a significant difference between musicians' and non-musicians' McGurk effect sizes.”

Again this is not correct, and, indeed, false. Proverbio et al. (2016) showed a significant decrease in the ability to correctly categorize phonemes in controls but NOT in musicians, and reported all the statistics!

<<as a function of the phonemes auditorily perceived showed the statistical significance of “condition” factor (congruent vs. incongruent conditions) ($F_{8,112} = 3.646$; $p < 0.0008$), but only for controls

as a function of the labial (lip movements) perceived was statistically significant ($F_{8,112} = 2.685$; $p < 0.0097$), but only for controls.>>

ABSTRACT. The authors state “and changing from a between- to a within-participants manipulation).!!

In Proverbio et al et al. 2016 the manipulation was WITHIN participants. ???

Introduction

The authors state: “People comprehend physical stimuli, they integrate information from multiple sensory modalities to generate a psychological percept. “

This is too generic, include appropriate scientific quotations relative to AV integration processes. Again please use a scientific language!

For example, the way people perceive a sound can be modulated by visual sensory information accompanying the sound. Perhaps the most famous example of this is the McGurk effect, whereby people tend to mis-perceive sounds.

This is incorrect. The McGurk effect has nothing to do with “sounds”, but with phoneme categorizations. The two processes are completely different (please read and include some literature)

Line 42 “experience changes some mechanisms of basic sound perception” again there are many evidences that musical ability modify linguistic perception and ability. We are not talking only about a finer acoustic ability

Page 2 (lines 47-51) the experiment did not reveal a significant difference between musicians' accuracy in audiovisual "McGurk" stimuli and audio-only stimuli, they nevertheless showed a large numerical effect in the direction of a typical McGurk effect (higher accuracy in audio-only than audiovisual stimuli).

This is profoundly incorrect and false, and should be imperatively changed!

In Proverbio et al. (2016) study, both musicians and controls showed no difference between the auditory unimodal and audiovisual congruent conditions

(Musicians: visual = 10%; auditory = 90%; audiovisual = 93.75%. Controls: visual = 11.25%; auditory = 93.75%; audiovisual = 95%)

The authors state on page 2 (lines 51-52) the experiment likely had low power to detect a significant effect, as this critical comparison was between only 10 participants who heard audio-only stimuli and 20 who saw audiovisual stimuli

That's totally incorrect and false. In Proverbio et al. (2016) say (right before the results)

"In this study, 80 healthy, age-matched, graduate male and female volunteers were tested both in multimodal (McGurk condition) and unimodal (only auditory and only visual) conditions."

Did you even read the paper?!

The authors state on line 27 page 4: "As Proverbio and colleagues had 30 musicians and 30 non-musicians"

This is again incorrect!

The stimuli used were different from Proverbio et al (2016)

The authors state: "we limited ourselves to consonants that form existing morphemes in both Mandarin and Cantonese in the frame /_a/. We did not use /n/ because /n/ and /l/ are merged in many southern dialects of Mandarin and Cantonese. We also did not use /g/ because, while both /kaɿ / and /gaɿ / are not very meaningful in Mandarin (they are mainly used in phonetic borrowings, like 咖啡/kaɿ fei ɿ / coffee"), /ka ɿ / is much more frequent."

The stimuli used by Stephen Politzer-Ahles & Lei Pan are full of acoustic noise, and this is absolutely not comparable to the experimental conditions used by Proverbio et al (2016) study in which there was silence in background! Phoneme perception is worsened by the noise presence.

Reviewer: 3

Comments to the Author(s)

The authors have presented a plan to replicate a recent paper that compared the McGurk effect between musicians and non-musicians in Italy. The methods are straightforward and I think the original manuscript is a good one to attempt to replicate.

I am overall in favor of this replication attempt, but think the authors should do a bit more work to emphasize the generalizability of the current findings. Specifically, a power analysis of sorts would help determine if even n=60 per group is adequate to find small effects. As the authors' are aware from the Gelman articles they cite, a problem with studying small effects with small sample sizes is that any statistically significant effect tends to be exaggerated -- see also [1] below for application of this idea to between-group studies of the McGurk effect. This is important to ensure that these replication results are more decisive.

Second, the authors note that the "ga" syllable was not used (or will not be used), because of the subjects' native language. References concerning the robustness of the McGurk effect in their chosen population should be included to address head on any possibility of floor effect hiding any potential group effect.

[1] Magnotti JF, Beauchamp MS (2018) Published estimates of group differences in multisensory integration are inflated. PLoS ONE 13(9): e0202908.

<https://doi.org/10.1371/journal.pone.0202908>

[2] Sekiyama, K (1997). Cultural and linguistic factors in audiovisual speech processing: The McGurk effect in Chinese subjects. *Perception & Psychophysics*, 59(1), 73-80.

[3] Chen TH, & Massaro DW (2004). Mandarin speech perception by ear and eye follows a universal principle. *Perception & psychophysics*, 66(5), 820-836.

[4] Magnotti JF, Mallick DB, Feng G, Zhou B, Zhou W, & Beauchamp, MS (2015). Similar frequency of the McGurk effect in large samples of native Mandarin Chinese and American English speakers. *Experimental brain research*, 233(9), 2581-2586.

Author's Response to Decision Letter for (RSOS-181868.R0)

See Appendix A.

RSOS-181868.R1 (Revision)

Review form: Reviewer 1 (Rebecca Hirst)

Do you have any ethical concerns with this paper?

No

Have you any concerns about statistical analyses in this paper?

No

Recommendation?

Accept in principle

Comments to the Author(s)

Please find comments in attached document (Appendix B).

Review form: Reviewer 2 (Alice Mado Proverbio)

Do you have any ethical concerns with this paper?

Yes

Have you any concerns about statistical analyses in this paper?

No

Recommendation?

Reject

Comments to the Author(s)

I am absolutely not satisfied with the manuscript, that has not been revised at all. The paper is full of imprecision and falsities, and the style is far from being scientific.

Decision letter (RSOS-181868.R1)

09-Jan-2019

Dear Dr Politzer-Ahles

On behalf of the Editor, I am pleased to inform you that your Manuscript RSOS-181868.R1 entitled "Are skilled musicians not subject to the McGurk effect?" has been accepted in principle for publication in Royal Society Open Science. The reviewers' and editors' comments are included at the end of this email.

You may now progress to Stage 2 and complete the study as approved.

You must now register your approved protocol on the Open Science Framework (<https://osf.io/rr>), either publicly or privately under embargo until submission of the Stage 2 manuscript. You should register the protocol even where the study has already been undertaken. Please note that a time-stamped, independent registration of the protocol is mandatory under journal policy, and manuscripts that do not conform to this requirement cannot be considered at Stage 2. The protocol should be registered unchanged from its current approved state. Please include a URL to the protocol in your Stage 2 manuscript.

When ready please resubmit your paper for peer review as a Stage 2 Replication. Please note that your manuscript can still be rejected for publication at Stage 2 if the Editors consider any of the following conditions to be met:

- The Introduction and methods deviated from the approved Stage 1 submission (required).
- The authors' conclusions were not considered justified given the data.

We encourage you to read the complete guidelines for authors concerning Stage 2 submissions at: <http://rsos.royalsocietypublishing.org/page/replication-studies#AuthorsGuidance>. Please especially note the requirements for data sharing and that withdrawing your manuscript will result in publication of a Withdrawn Registration.

Once again, thank you for submitting your manuscript to Royal Society Open Science and I look forward to receiving your Stage 2 submission. If you have any questions at all, please do not hesitate to get in touch. We look forward to hearing from you shortly with the anticipated submission date for your stage two manuscript.

Kind regards,
Professor Chris Chambers
Royal Society Open Science
openscience@royalsociety.org

on behalf of Chris Chambers (Registered Reports Editor, Royal Society Open Science)
openscience@royalsociety.org

Editor Comments to Author (Professor Chris Chambers):

The revised manuscript was returned to two of the three original reviewers who assessed the initial Stage 1 submission (Reviewers 1 and 2). Reviewer 1 is now satisfied with the manuscript and recommends IPA (though please note the reviewer's point about noise in the stimuli for the Discussion at Stage 2). Reviewer 2 remains unsatisfied and recommends rejection.

The reviews are therefore strongly polarised, with Reviewers 1 and 3 in favour of IPA and Reviewer 2 against. Having read your submission, your response to Reviewer 2, and the original target paper for the replication, I am satisfied that you have adequately rebutted the concerns of Reviewer 2. Stage 1 IPA is therefore granted.

Reviewers' comments to Author:

Reviewer: 1

Comments to the Author(s)

Please find comments in attached document.

Reviewer: 2

Comments to the Author(s)

I am absolutely not satisfied with the manuscript, that has not been revised at all.

The paper is full of imprecision and falsities, and the style is far from being scientific.

Author's Response to Decision Letter for (RSOS-181868.R1)

See Appendix C.

RSOS-181868.R2 (Revision)

Review form: Reviewer 1 (Rebecca Hirst)

Do you have any ethical concerns with this paper?

No

Have you any concerns about statistical analyses in this paper?

No

Recommendation?

Accept with minor revision

Comments to the Author(s)

This is the third time I have read this manuscript (although the first time I have read the results and discussion). The manuscript reports an attempted replication of Proverbio et al (2016), reporting that Musicians are not susceptible to the McGurk effect. The results suggest that

Musicians are in fact susceptible to the effect, perhaps more so than non-musicians. I found the manuscript and the available data clear and easy to navigate. I do have a few comments on the results/discussion:

1. Although I am aware that this was not in the pre-registered analysis plan, it is curious that the two coding conditions are not statistically compared - such that it could be said that including stimuli with inconsistent place articulation results in significantly lower accuracy than consistent place articulation? Even if this is not explored statistically, it seems this could be given some mention in the discussion. This is important to guide future research design, and also because the authors suggest that lower McGurk effects in Proverbio et al's study might in part have been due to the type of stimuli included.

2. Page 12, line 34 "If anything, is musicians are better than non-musicians at processing speech in noise" another very relevant reference here is Coffey et al "Speech-in-noise perception in musicians: A review"

3. Page 13 line 4 "The present study obviously rules out those conclusions: if musicians are indeed subject to the McGurk effect, then McGurk studies do not provide any evidence for brain-level functional reorganization as a result of musical training." This seems quite a bold claim considering this study still reported between groups differences (albeit in the opposite direction to Proverbio and colleagues!). Differences between groups surely suggests that the McGurk effects might show some functional differences - although the exact nature of these would need brain imaging studies to confirm the nature of plasticity (which the authors acknowledge this study does not rule out).

Review form: Reviewer 3

Do you have any ethical concerns with this paper?

No

Have you any concerns about statistical analyses in this paper?

No

Recommendation?

Accept with minor revision

Comments to the Author(s)

The authors have revealed the results of their replication attempt. They find substantial McGurk effect in musicians and non-musicians, contra the previous report. The increased sample size, suitable and pre-registered analysis plan, and robust experimental design provide confidence that these results easily meet the requirements for a solid replication study. Because the results are not in-line with the past paper, there will likely be substantial push back from any potential strong adherents to the original result. For this reason, and to make the paper more suitable to a general audience, I've suggested several changes below that will make the report easier to understand. Most importantly, a table that directly addresses points of similarity and dissimilarity between the papers is essential prior to publication.

Results

Page 9, bottom paragraph

The authors provide two comparisons between musicians and non-musician levels of mcgurk. CI says no diff, GLMM says yes, likely b/c of odds-ratios are more sensitive to changes in probability scale. It would be helpful for a non-specialist reader to have this apparent (not actual)

contradiction explained away.

Additionally, the authors should make clear which parameters are being referred to when presenting the coefficients from the GLMM. For instance, in the sentence, "The statistical analysis confirmed that musicians have a highly significant McGurk effect ($b=-2.34$, $z=-18.37$, $p<.001$), and that it in fact is significantly larger than the non-musicians' McGurk effect ($b=0.46$, $z=2.72$, $p=.007$)" I would think the first parameter is the main effect (intercept term) of condition (Incong < audio) and the second parameter is the interaction between condition and group, but this should be made explicit. Similar for secondary analysis, page 10, lines 38-42.

Discussion

The authors should remind the reader of the relative sample sizes in the first sentence, rather than just referring to the proportional increase.

I found the first paragraph of the discussion hard to follow as it tallies similarities and differences between the results. A figure here with appropriate confidence intervals would bring great clarity and provide a useful summary of the replication attempt.

The second paragraph also devolves a bit into diatribe listing the problems with the original study's numerical analysis. I think this is a serious issue, and thus should not be relegated to a, quite long, 79-word parenthetical remark. Instead, the authors should create separate paragraphs for each reason a direct comparison of the numerical results is difficult/impossible.

A rehash of major differences between the original and replication study would be helpful, perhaps as a table. This allows readers to form their own opinion about how closely the replication followed the original. I think the point about musicians (in general) perceiving the McGurk illusion is well-demonstrated by this new paper, but perhaps there were things in the original study that made a null-finding more likely, e.g., sample/population characteristics, stimuli used etc. Again, having this information in a table will facilitate easy comparison and provide some structure for the discussion.

Minor Points

Page 12, line 8, change "minus" to "vs."

Page 12, line 16-17 should be made stronger than just "we believe our result... to be the more robust one." Indeed, all the improvements listed render the new results simply more robust.

Figure 1. The data are plotted as proportion (0.0 to 1.0), but referred to as percentages (0 to 100%). Probably easiest to convert the graph to percentages.

Decision letter (RSOS-181868.R2)

05-Feb-2019

Dear Dr Politzer-Ahles

On behalf of the Editor, I am pleased to inform you that your Stage 2 Replication submission RSOS-181868.R2 entitled "Skilled musicians are indeed subject to the McGurk effect" has been accepted for publication in Royal Society Open Science subject to minor revision in accordance with the referee suggestions. Please find the referees' comments at the end of this email.

The reviewers and Subject Editor have recommended publication, but also suggest some minor revisions to your manuscript. Therefore, I invite you to respond to the comments and revise your manuscript.

Please also ensure that all the below editorial sections are included where appropriate (a non-exhaustive example is included in an attachment):

- Ethics statement

- Data accessibility

If you wish to submit your supporting data or code to Dryad (<http://datadryad.org/>), or modify your current submission to dryad, please use the following link:
<http://datadryad.org/submit?journalID=RSOS&manu=RSOS-181868.R2>

- Competing interests

- Authors' contributions

- Acknowledgements

- Funding statement

Because the schedule for publication is very tight, it is a condition of publication that you submit the revised version of your manuscript within 7 days (i.e. by the 13-Feb-2019). If you do not think you will be able to meet this date please let me know immediately.

- 1) A text file of the manuscript (tex, txt, rtf, docx or doc), references, tables (including captions) and figure captions. Do not upload a PDF as your "Main Document".
- 2) A separate electronic file of each figure (EPS or print-quality PDF preferred (either format should be produced directly from original creation package), or original software format)
- 3) Included a 100 word media summary of your paper when requested at submission. Please ensure you have entered correct contact details (email, institution and telephone) in your user account
- 4) Included the raw data to support the claims made in your paper. You can either include your data as electronic supplementary material or upload to a repository and include the relevant DOI within your manuscript
- 5) Included your supplementary files in a format you are happy with (no line numbers, Vancouver referencing, track changes removed etc) as these files will NOT be edited in production

Kind regards,
Professor Chris Chambers
Royal Society Open Science
openscience@royalsociety.org

on behalf of Chris Chambers (Registered Reports Editor, Royal Society Open Science)
openscience@royalsociety.org

Associate Editor Comments to Author (Professor Chris Chambers):

The manuscript was returned to two of the three reviewers who assessed the protocol at Stage 1. Both reviewers are positive about the Stage 2 submission and recommend publication following minor revision, focusing on clarifications to the presentation, including the interpretation of the results and the Discussion. Concerning point #1 from Reviewer 1, it would be acceptable for the authors to report an additional analysis if it is noted that this analysis was undertaken in addition

to the approved Stage 1 plan. Provided the authors address the reviewers' comments thoroughly in revision, final acceptance should be forthcoming without requiring further in-depth review.

Reviewers' comments to Author:

Reviewer: 3

Comments to the Author(s)

The authors have revealed the results of their replication attempt. They find substantial McGurk effect in musicians and non-musicians, contra the previous report. The increased sample size, suitable and pre-registered analysis plan, and robust experimental design provide confidence that these results easily meet the requirements for a solid replication study. Because the results are not in-line with the past paper, there will likely be substantial push back from any potential strong adherents to the original result. For this reason, and to make the paper more suitable to a general audience, I've suggested several changes below that will make the report easier to understand. Most importantly, a table that directly addresses points of similarity and dissimilarity between the papers is essential prior to publication.

Results

Page 9, bottom paragraph

The authors provide two comparisons between musicians and non-musician levels of mcgurk. CI says no diff, GLMM says yes, likely b/c of odds-ratios are more sensitive to changes in probability scale. It would be helpful for a non-specialist reader to have this apparent (not actual) contradiction explained away.

Additionally, the authors should make clear which parameters are being referred to when presenting the coefficients from the GLMM. For instance, in the sentence, "The statistical analysis confirmed that musicians have a highly significant McGurk effect ($b=-2.34$, $z=-18.37$, $p<.001$), and that it in fact is significantly larger than the non-musicians' McGurk effect ($b=0.46$, $z=2.72$, $p=.007$)" I would think the first parameter is the main effect (intercept term) of condition (Incong < audio) and the second parameter is the interaction between condition and group, but this should be made explicit. Similar for secondary analysis, page 10, lines 38-42.

Discussion

The authors should remind the reader of the relative sample sizes in the first sentence, rather than just referring to the proportional increase.

I found the first paragraph of the discussion hard to follow as it tallies similarities and differences between the results. A figure here with appropriate confidence intervals would bring great clarity and provide a useful summary of the replication attempt.

The second paragraph also devolves a bit into diatribe listing the problems with the original study's numerical analysis. I think this is a serious issue, and thus should not be relegated to a, quite long, 79-word parenthetical remark. Instead, the authors should create separate paragraphs for each reason a direct comparison of the numerical results is difficult/impossible.

A rehash of major differences between the original and replication study would be helpful, perhaps as a table. This allows readers to form their own opinion about how closely the replication followed the original. I think the point about musicians (in general) perceiving the McGurk illusion is well-demonstrated by this new paper, but perhaps there were things in the original study that made a null-finding more likely, e.g., sample/population characteristics, stimuli used etc. Again, having this information in a table will facilitate easy comparison and provide some structure for the discussion.

Minor Points

Page 12, line 8, change "minus" to "vs."

Page 12, line 16-17 should be made stronger than just "we believe our result... to be the more robust one." Indeed, all the improvements listed render the new results simply more robust.

Figure 1. The data are plotted as proportion (0.0 to 1.0), but referred to as percentages (0 to 100%). Probably easiest to convert the graph to percentages.

Reviewer: 1

Comments to the Author(s)

This is the third time I have read this manuscript (although the first time I have read the results and discussion). The manuscript reports an attempted replication of Proverbio et al (2016), reporting that Musicians are not susceptible to the McGurk effect. The results suggest that Musicians are in fact susceptible to the effect, perhaps more so than non-musicians. I found the manuscript and the available data clear and easy to navigate. I do have a few comments on the results/discussion:

1. Although I am aware that this was not in the pre-registered analysis plan, it is curious that the two coding conditions are not statistically compared - such that it could be said that including stimuli with inconsistent place articulation results in significantly lower accuracy than consistent place articulation? Even if this is not explored statistically, it seems this could be given some mention in the discussion. This is important to guide future research design, and also because the authors suggest that lower McGurk effects in Proverbio et al's study might in part have been due to the type of stimuli included.
2. Page 12, line 34 "If anything, is musicians are better than non-musicians at processing speech in noise" another very relevant reference here is Coffey et al "Speech-in-noise perception in musicians: A review"
3. Page 13 line 4 "The present study obviously rules out those conclusions: if musicians are indeed subject to the McGurk effect, then McGurk studies do not provide any evidence for brain-level functional reorganization as a result of musical training." This seems quite a bold claim considering this study still reported between groups differences (albeit in the opposite direction to Proverbio and colleagues!). Differences between groups surely suggests that the McGurk effects might show some functional differences - although the exact nature of these would need brain imaging studies to confirm the nature of plasticity (which the authors acknowledge this study does not rule out).

Author's Response to Decision Letter for (RSOS-181868.R2)

See Appendix D.

Decision letter (RSOS-181868.R3)

07-Feb-2019

Dear Dr Politzer-Ahles:

It is a pleasure to accept your Stage 2 Replication entitled "Skilled musicians are indeed subject to the McGurk effect" in its current form for publication in Royal Society Open Science. Congratulations on being the first fully accepted Replication article since we launched this new format.

on behalf of Professor Chris Chambers (Subject Editor)
openscience@royalsociety.org

Appendix A

Reviewer 1

For the analyses in this paper the McGurk effect is quantified as the difference between incongruent and auditory only. However, it looks as though Proverbio et al also compared congruent and incongruent trials (in addition to incongruent vs auditory only). A short justification as to why this effect was not explored in this study could make it easier to compare between studies.

Their key analysis on which they base the claim that skilled musicians do not show the McGurk effect is based on just comparing incongruent and auditory only; the congruent audiovisual trials only seem to be used in follow-up analyses breaking down the effect in different phonemes (although I don't see any explanation of why they switched from one comparison to the other). We have added the following parenthetical in the main text to explain this: "*(Proverbio and colleagues¹² key analysis comparing musicians and non-musicians is based on comparing audio-only and audiovisual incongruent conditions, as shown in their Figure 1 and the first paragraph of their Results section. The only time they use the audiovisual congruent condition to quantify McGurk effects is in follow-up analyses examining interactions with different phonemes [their Figures 2 and 3].)*".

I was unable to open the .zil data files on my mac operating system. Perhaps providing data in a second format (or providing info on how to open the .zil files - even if it means the user should find a windows computer) would increase data accessibility further (although another format may not work with the available R scripts?).

Sorry for this confusion. We have added some extra instruction on the OSF page now (under the "wiki" section) explaining several ways .zil files can be opened. We have also added a .csv file that has all the results compiled in one place; hopefully this will make it easier to access the data.

Reviewer 2

The authors stat that "sounds are often mis-perceived" in McGurk effect. This is incorrect. Sounds are perceived fine, phonemes are mis-catagorized

As this issue is immaterial to the claims at stake in the paper (we are only interested in whether or not the McGurk effect is different between musicians and non-musicians, and have no commitment to any particular theoretical description of it or its locus), we are happy to change the way this background information is described. We have done so throughout the new manuscript, rewording these parts to present the McGurk effect as something specifically about the categorization of speech sounds; these are indicated with Track Changes in the revised manuscript. (We have not used the term "phoneme", as a phoneme is not the same as a speech sound; multiple speech sounds can be realizations of the same phoneme).

The authors state in the abstract:

"it is not clear, however, if this is intended to mean that skilled musicians do not experience the McGurk effect at all, or if they just experience it to a lesser magnitude than non-musicians"

This is not scientific language!! The paper by Proverbio et al. (2016) showed that:

<<The analyses of data performed as a function of the phonemes auditorily perceived showed the statistical significance of "condition" factor (congruent vs. incongruent conditions) ($F_{8,112} = 3.646$; $p < 0.0008$), but only for controls. The analysis of group effects and Tukey post-hoc tests showed no significant difference between the congruent and the McGurk conditions in musicians but only in controls.>> In scientific terms: there was no difference in the ability to correctly categorize phonemes in the AV congruent vs incongruent condition (meaning at all). The tendencies visible in the figure but not statistically significant cannot be considered as "to a lesser magnitude", this is not the scientific methods

If I have understood this comment correctly, the reviewer is saying that we should not claim that musicians showed a numerical pattern in the direction of a McGurk effect in Proverbio et al. (2016) because that pattern was not significant.

Simply put, this is not how statistics works. As we all learn in introductory statistics courses, failure to reject the null hypothesis does not license acceptance of the null hypothesis. In the context of a McGurk experiment, what that means is: the fact that some group is not *significantly* less accurate on incongruent than other conditions does not mean that their accuracies are exactly the same. Lack of statistical significance does not allow one to include that there is no difference "at all" (R2's words). The authors seem to be thinking in terms of old-style Neyman-Pearson decision theory, whereby $p=.05$ is an absolute cutoff, and indeed many people still do attempt to operate this way (see review in, e.g., Gigerenzer, 2004), but that is only a framework for how to decide what to do on the basis of statistics (i.e., to decide "my theory was supported" or "my theory was not supported"); it does not mean that we close our eyes and pretend that a real observed numerical difference is suddenly not there. An observed value is an observed value, no matter what its associated p -value is.

The reviewer also seems to have misunderstood what we mean by "magnitude". Magnitude refers the raw size of an effect (in whatever units make sense for measuring this effect), not to its statistical significance. Let me contextualize this by giving an example in McGurk terms. Imagine we have two groups (see example image to the right), and Group A is 10 percentage points less accurate on identification of McGurk stimuli (audiovisual incongruent) than audio-only stimuli, and Group B is 20 points less accurate. Further, imagine that Group A's 10-point McGurk effect is statistically significant (as indicated in the figure by the confidence interval that does not include zero),

whereas Group B's is not (perhaps Group B has a much smaller number of participants, or the effect has a very high standard deviation around 20; those are two reasons that a numerically large effect might not be statistically significant). We would still say that Group B's effect had a larger *magnitude*, even though it is less significant. Magnitude and significance are not the same thing. See Gelman and Carlin (<http://www.stat.columbia.edu/~gelman/research/published/retropower20.pdf>) for more discussion of magnitude and magnitude errors.

By this token, everything already in the current manuscript is reasonable. It is indeed true that the question of whether musicians' McGurk effect is 0 is a different question than that of whether musician's McGurk effect is smaller than non-musician's McGurk effect. (To put it in statistical terms: one potential H_0 is $McGurk_{musician} = 0$, and another possible H_0 is $McGurk_{nonmusician} - McGurk_{musician} = 0$; these are different hypotheses which need different tests, and it is not clear which one is the most relevant for testing the theoretical claims made by Proverbio et al., 2016.) It is also true that in their experiment (particularly, in Figure 1, which I reproduce to the right for ease of exposition) musicians show a numerical but non-significant McGurk effect, which is smaller in magnitude than non-musicians' McGurk effect but which is bigger than zero, and which is not significantly different from non-musicians' McGurk effect (Proverbio and colleagues' words: "No interaction between group and condition was found for this contrast."). In short, there is no need to change anything in the manuscript in response to this comment from the reviewer, because none of the parts pointed out are actually incorrect, and the reviewer's concern appears to stem from a misunderstanding of statistics.

The authors state: "The study also does not statistically demonstrate either of these conclusions" This is totally false, as explained below, and should be imperatively changed!

The two possible conclusions the reviewer is referring to are (1) that musicians' McGurk effect is smaller than non-musicians', and (2) that musicians' McGurk effect is zero. We are correct that the study by Proverbio and colleagues (2016) does not statistically demonstrate these. Regarding #1, their study does not report a significant interaction or difference-of-differences to show that musicians have a smaller McGurk effect than non-musicians; rather, the crucial interaction is non-significant, as shown in the sentence quoted above in our response to the previous comment. The only statistical evidence they have for the claim that musicians' McGurk effect is smaller is that non-musicians have a significant McGurk effect (when "McGurk effect" is quantified by comparing the congruent and incongruent trials, rather than comparing the audio-only and incongruent trials as is shown in Figure 1) whereas non-musicians do not. But this does not prove that their effects are significantly different. Our manuscript already cites Gelman & Stern (2006), titled "The difference between 'significant' and 'not significant' is not itself statistically significant". Here is, verbatim, what our manuscript already says: *"Furthermore, the experiment also tested a control group of participants without musical experience, and there was not a significant interaction between the groups and the type of stimuli perceived. Without a significant interaction, the conclusion that non-musicians had a McGurk effect and musicians did not is not necessarily justified: the fact that one group shows a significant effect and another group does not show a significant effect is not in of itself sufficient evidence that the two groups are significantly different from one another³."* I don't know what more can be said, as this is already quite clear.

Regarding the second possible conclusion, this is also not statistically demonstrated in the Proverbio et al. (2016) paper. As we have explained in our response to the previous comment above, the fact that an effect is not statistically significant does not mean that the effect is zero. There are methods for getting statistical evidence to support the claim that an effect is probably zero (equivalence hypothesis tests,

Bayes factors, etc.) but none of those were carried out in that paper.

The authors state: "as it does report a numerical (albeit non-significant) McGurk effect for musicians, and does not report a significant difference between musicians' and non-musicians' McGurk effect sizes."

Again this is not correct, and, indeed, false. Proverbio et al. (2016) showed a significant decrease in the ability to correctly categorize phonemes in controls but NOT in musicians, and reported all the statistics!

As discussed in our responses to the other comments above, the reviewer's concern here is unfounded because it is based on a misunderstanding of how statistics works. What we said there (that musicians have a non-zero McGurk effect, which is neither significantly different from zero nor significantly different from non-musicians' McGurk effect) is not at all inconsistent with the fact that Proverbio et al. (2016) showed a significant McGurk effect in non-musicians but not musicians.

ABSTRACT. The authors state "and changing from a between- to a within-participants manipulation).!!

In Proverbio et al et al. 2016 the manipulation was WITHIN participants. ???

It was not within participants. Here let me explain how I know that.

First we need to know what "the manipulation" was. A McGurk effect could be quantified by comparing performance on audiovisual incongruent trials to performance on audio-only trials, or to performance on audiovisual congruent trials. In the study by Proverbio and colleagues (2016) the critical claim (that musicians don't have a McGurk effect) seems to be based on the former comparison: that is what they show in their Figure 1, and that is the comparison they use for the most direct analysis. (They switch to using the latter comparison when they start breaking down these comparisons

within each phoneme; for our study, we had no hypotheses about interactions with phoneme, so we ignore this.) Therefore, for the purposes relevant to this discussion, "the manipulation" is the comparison between audio-only and audiovisual incongruent stimuli.

Now, was that manipulated between or within participants? Before the Results section, Proverbio and colleagues (2016) state, "*In this study, 80 healthy, age-matched, graduate male and female volunteers were tested both in multimodal (McGurk condition) and unimodal (only auditory and only visual) conditions*". This sentence is semantically ambiguous: it could mean that each of these people participated in each condition (a distributive reading) or it could mean each of these conditions was participated in but not necessarily by the same people (with 80 unique people in one condition and 80 unique people in the other, or with $x_{\{x < 80\}}$ unique people in one condition and $80-x$ in the other). More unambiguous is a statement the authors make in the Stimuli section: "*To avoid an excessive time length of the experimental procedure each subject was presented with 64 videos in the McGurk condition (40 Ss), or to the unimodal auditory condition (20 Ss), or to the unimodal visual condition (20 Ss)*." This unambiguously states that the people who participated in the audio-only condition were not the same people who participated in the audiovisual incongruent condition; i.e., it was a between-participants manipulation.

We don't need to just guess from the writing; we can also see the data. Below, I reproduce for ease of exposition Proverbio and colleagues' (2016) Table 1, showing the response proportions for musicians in the audiovisual stimulation conditions. It is clear that the percentages in every cell are multiples of 5, which means there were 20 observations in each cell. (The chance that there were 40 observations per cell but never any "2.5%" sort of responses is infinitesimally small.) This means there were 20 musicians, not 40, in this stimulation condition. In other words, not all musicians participated in the same stimulation condition; in other words, this is a between-participants manipulation.

		VISUAL INPUT						
		LA	DA	TA	GA	KA	NA	BA
AUDITORY INPUT	LA	La = 100	La = 95 Bla = 5	La = 100	La = 100	La = 100	La = 100	La = 80 Pla = 15 Mla = 5
	DA	Da = 95 Lda = 5	Da = 90 Bda = 10	Da = 100	Da = 90 Bda = 10	Da = 100	Da = 100	Da = 85 Bda = 15
	TA	Ta = 100	Ta = 100	Ta = 100	Ta = 95 Mta = 5	Ta = 95 Lta = 5	Ta = 95 Pta = 5	Ta = 90 Pta = 10
	GA	Ga = 90 Lga = 10	Ga = 95 Bka = 5	Ga = 95 Dga = 5	Ga = 95 Nga = 5	Ga = 95 Dga = 5	Ga = 65 Da = 20 Pga = 5 Gna = 5 Pda = 5	Ga = 90 Mga = 5 Pga = 5
	KA	Ka = 95 Lka = 5	Ka = 100	Ka = 95 Dka = 5	Ka = 95 Tka = 5	Ka = 95 Lka = 5	Ka = 95 Lka = 5	Ka = 90 Pka = 10
	NA	Na = 100	Na = 95 Mna = 5	Na = 100	Na = 100	Na = 95 Mna = 5	Na = 95 Mna = 5	Na = 85 Mna = 10 Pna = 5
	BA	Ba = 85 Lba = 10 Da = 5	Ba = 95 Dba = 5	Ba = 75 Da = 20 Dba = 5	Ba = 100	Ba = 100	Ba = 90 Bam = 10	Ba = 100
	PA	Pa = 50 Ta = 40 Lpa = 5 Lta = 5	Pa = 100	Pa = 70 Ta = 20 Tpa = 5 Pan = 5	Pa = 70 Ba = 10 La = 5 Pga = 5 Ta = 5 A = 5	Pa = 65 Ta = 35	Pa = 95 Lpa = 5	Pa = 100

Table 1. MUSICIANS: Qualitative description of auditory percepts recorded in the MGurk experiment as a function of phonetic (left) and labial (top) inputs. In each box the percentages (%) of musicians reporting given percept are displayed.

The authors state: "People comprehend physical stimuli, they integrate information from multiple sensory modalities to generate a psychological percept. " This is too generic, include appropriate scientific quotations relative to AV integration processes. Again please use a scientific language!

We consider a detailed theoretical literature review about the McGurk effect to be beyond the scope of this short paper, which is just a methodologically rigorous replication study. In any case, even if the statement that the reviewers have quoted here is general, the reviewer has not argued that it is incorrect. Indeed, it is the first sentence in the body of the text, so of course it is general; most articles start with a statement of a general phenomenon or issue and then work down to the specifics, and ours is no exception. The reviewer has also not explained what about this statement is not "scientific" (other than the fact that it's not longer or more

detailed or including more references; none of these are generally accepted as criteria for letting something qualify as "scientific"). Therefore, we have opted not to change it.

For example, the way people perceive a sound can be modulated by visual sensory information accompanying the sound. Perhaps the most famous example of this is the McGurk effect, whereby people tend to mis-perceive sounds.

This is incorrect. The McGurk effect has nothing to do with "sounds", but with phoneme categorizations. The two processes are completely different (please read and include some literature)

This issue was raised in Reviewer 2's first comment. As we indicated in our response to that comment, we have edited the wording about this issue throughout the manuscript. This particular sentence (the first paragraph of the comment copied above is a quotation from our manuscript) is one of the ones we have edited; in the revised manuscript, we now say "whereby people tend to mis-perceive speech sounds", rather than just sounds.

Line 42 "experience changes some mechanisms of basic sound perception" again there are many evidences that musical ability modify linguistic perception and ability. We are not talking only about a finer acoustic ability

This we have also changed; see the above comment. We replaced "basic sound perception" with "speech perception".

Page 2 (lines 47-51) the experiment did not reveal a significant difference between musicians' accuracy in audiovisual "McGurk" stimuli and audio-only stimuli, they nevertheless showed a large numerical effect in the direction of a typical McGurk effect (higher accuracy in audio-only than audiovisual stimuli).

This is profoundly incorrect and false, and should be imperatively changed!

In Proverbio et al. (2016) study, both musicians and controls showed no difference between the auditory unimodal and audiovisual congruent conditions

(Musicians: visual = 10%; auditory = 90%; audiovisual = 93.75%. Controls: visual = 11.25%; auditory = 93.75%; audiovisual = 95%)

The reviewer has misunderstood our statement. We are talking about the comparison between audio-only and audiovisual *incongruent* stimuli; audiovisual congruent stimuli are irrelevant here. We have added the word "incongruent" in the parenthetical there to make this clear. Proverbio and colleagues' (2016) musicians did indeed show a numerical McGurk effect in this comparison, as shown in their Figure 1 (copied above).

The authors state on page 2 (lines 51-52) the experiment likely had low power to detect a significant effect, as this critical comparison was between only 10 participants who heard audio-only stimuli and 20 who saw audiovisual stimuli

That's' totally incorrect and false. In Proverbio et al. (2016) say (right before the results) "In this study, 80 healthy, age-matched, graduate male and female volunteers were tested both in multimodal (McGurk condition) and unimodal (only auditory and only visual) conditions."

Did you even read the paper?!

See our response a few comments above. Proverbio and colleagues' (2016) paper had 20 musicians in the audiovisual condition, 10 in the audio condition, 10 in the visual condition, and likewise 20 nonmusicians in the audiovisual condition, 10 in audio, and 10 in visual.

The authors state on line 27 page 4: "As Proverbio and colleagues had 30 musicians and 30 non-musicians"

This is again incorrect!

The reviewer has mis-quoted us. The relevant part they are quoting from is: "As Proverbio and colleagues² had 30 musicians and 30 non-musicians **in the critical conditions**"

(emphasis added). The "critical conditions" are audio-only and audiovisual stimulation; the visual stimulation condition is not relevant to any of the claims made. In the audio-only and audiovisual stimulation conditions, they did indeed have 30 musicians and 30 non-musicians, as explained in our comment directly above this one. Here it is laid out in a table:

	Audiovisual	Audio	Visual
Musicians	20	10	10
Non-musicians	20	10	10

These are the numbers of participants in Proverbio and colleagues' (2016) design, as reported in their Methods section (specifically, under "Stimuli") and as reflected in Tables 1 and 2 of their paper (see our discussion of their Table 1 a few comments above).

The stimuli used by Stephen Politzer-Ahles & Lei Pan are full of acoustic noise, and this is absolutely not comparable to the experimental conditions used by Proverbio et al (2016) study in which there was silence in background! Phoneme perception is worsened by the noise presence.

The difference between our stimuli is something we intend to discuss in the Discussion section of the paper. For now, a few points bear mentioning:

- The presence of noise should not be relevant to the possibility of replicating the effect, as if noise worsens phoneme perception in one condition it should also do so in all the others. The only way it would have a different effect in the audiovisual incongruent condition is through the McGurk effect. Thus, the presence of noise does not confound the results. And the noise is the same for musicians and non-musicians, so it does not confound the comparison between these.
- The paper by Proverbio and colleagues (2016) does not make any mention of noise levels in the methods section; thus, the paper is not making any claims about

the relevance of this factor to the experiment. Of course any replication study will have small differences with the original (things like what kind of room it is conducted in, etc.). Those differences are not considered meaningful unless there was an *a priori* claim that the original study's effects would only hold under these certain kinds of conditions. If the ability to observe an effect depends on adherence to some certain experimental conditions that were never mentioned (so-called "hidden moderators", <http://datacolada.org/63>), this is something that needs to be discovered. In any case, if our results replicate Proverbio and colleagues' (2016), this will suggest that noise level is not a big deal for this effect. If they do not replicate Proverbio and colleagues' (2016), this will mean that their claim was incorrect or needs to be refined: either skilled musicians *are* indeed subject to the McGurk effect, or the claim needs to be sharpened to something like "Skilled musicians are not subject to the McGurk effect *unless there is sufficient noise*". In any case, that would be important information to know.

Reviewer 3

Specifically, a power analysis of sorts would help determine if even $n=60$ per group is adequate to find small effects. As the authors' are aware from the Gelman articles they cite, a problem with studying small effects with small sample sizes is that any statistically significant effect tends to be exaggerated -- see also [1] below for application of this idea to between-group studies of the McGurk effect. This is important to ensure that these replication results are more decisive.

We would like to have performed a power analysis, but this was not possible as the effect size from the original study is not known. (This is hinted in the manuscript but we did not draw much attention to it: the last "limitation" we point out about the study in the Introduction is that their analysis is not replicable. We did not want to spend a lot of time harping on it in this short paper, but the details are all publicly available in

the online comments to that article. In short, the results shown in their tables do not match the numbers shown in their figures or in their prose, and it is not clear how the values were transformed.) Without knowing that, we did not consider power analysis meaningful. We would like to have registered a "small telescopes" kind of analysis to judge whether this study's results are consistent with the original study, but given that we pre-registered a mixed-effects regression analysis and the original study uses ANOVA based on inscrutable values that cannot be reproduced, we did not see any meaningful way to calculate power or a $d_{33\%}$ effect size; we could of course do some procedures to get some numbers but we don't believe they would be meaningful, given all the unknowns still at play. We figure this study will be a step in the right direction because with our openly available data, people will be able to analyze the results any way they want and get estimates of the effect size and thus be able to perform power calculations for their future studies.

Without knowing exact power, the best we can do is do things we know will increase power. We know that, all else being equal, increasing N will increase power; thus, we doubled the sample size. We also know that, all else being equal, increasing the precision of the estimate will increase the power; that is why we pre-registered a targeted "McGurk" analysis that focuses only on the stimuli where McGurk effects are expected, because we figured this will get a more precise estimate of the effect we are looking at and thus will increase our power. We also use mixed effect logistic regression, which is a more appropriate statistical model for this kind of data and thus should offer more power than arcsin+ANOVA. Finally, the last way to increase power is to increase the magnitude of the effect. We didn't do anything specifically designed for this, and I guess it's an open question whether any details of our experiment design (using Chinese speakers, the way we recorded stimuli, the way we presented the experiment, etc.) would increase or decrease the magnitude of the effect. But we are at least confident that our experiment has higher power than the original (even if we don't know its exact power), given that we improved 3 out of the 4 things that could affect power. We have now updated the manuscript such

that the list of 5 changes in the Introduction section explicitly states how these should increase power.

We would like to thank the reviewer for providing useful references related to this topic. We have integrated them into the revised version of the manuscript.

Second, the authors note that the "ga" syllable was not used (or will not be used), because of the subjects' native language. References concerning the robustness of the McGurk effect in their chosen population should be included to address head on any possibility of floor effect hiding any potential group effect.

We would like to thank the reviewer for raising this issue. We have added a paragraph in the "Participants" section addressing the language issue. In short, we agree with the recent literature that Chinese speakers probably do not necessarily have a different McGurk effect than speakers of other language, and even if they do, that biases our results in favor of not finding a McGurk effect in musicians (i.e., in favor of supporting the original study). So if the results come out not supporting the original study, the choice of participant population would not be a confound.

Appendix B

Are skilled musicians not subject to the McGurk effect?

This is the second time I have reviewed this paper, and I believe that it is well written and technically sound in its analyses. The reviewers addressed my concerns regarding analytical comparisons and accessibility of data. I have a few remaining minor comments regarding the documentation of OSF files and theoretical considerations that may be considered in the discussion:

1. The .csv document is easier to access and interpret. However, a "ReadMe" doc defining each of the variables in the .csv file would be helpful. For example – what does "vidPoA" stand for? In turn, what do "velar" and "labial" ect represent? I realise that the authors have already provided an explanation to some extent in the wiki, however, this could be clearer (plus having the "ReadMe" doc in the data folder may make it more obvious where this is stored).
2. Although this is not in response to my comment, the author's response to Reviewer 2 regarding the noise in stimuli may need further. Specifically, the authors state:

"Thus, the presence of noise does not confound the results. And the noise is the same for musicians and non- musicians, so it does not confound the comparison between these ."

Although this is true, there are theoretical hypotheses that need to be considered. Specifically, if musicians are less susceptible to the McGurk effect due to some form of "auditory dominance" and sensory dominance is considered as an ability to detect signal in noise within the dominant modality, it might be that musicians actually detect the sound more clearly within acoustic noise relative to controls. This hypothesis is discussed briefly in the discussion of Hirst et al 2018 "The threshold for the McGurk effect in audio-visual noise decreases with development", in which it is stated "dominance may map onto an ability to identify a relevant signal (i.e. speech sound or lip movement) within the dominant modality rather than general susceptibility to noise in that modality.". Based on this, it would be predicted

that musicians “perceive” the speech sound more clearly in noise, and therefore rely less on visual signals to guide their perception – basically, they should be less influenced by visual information, and show smaller McGurk effects. As such, if acoustic noise is present, it is possible that this would actually inflate differences between musicians and nonmusicians. Although I believe this matter is less central to the core claims tested in this paper, if the authors intend to claim in the discussion that noise in stimuli would not have influenced between group differences, this is something that could be considered.

3. Page 3 line 42:

“An exact power analysis was not possible, as the exact effect size and variance structure in Proverbio and colleagues'² data is unknown and their results not reproducible” (comma after “possible”).

Appendix C

Dear Dr. Chambers,

Thank you for the opportunity to submit a Stage 2 manuscript. We would also like to express our thanks to all reviewers for taking the time to review the paper, and to Reviewer 1 for their new suggestions. We have taken R1's suggestions and added a README.txt file to OSF, added a paragraph discussing the background noise issue (towards the end of the Discussion), and making the typographical change. Our submission includes a Stage 2 manuscript with Track Changes to show the new additions with the results.

Best,

Steve Politzer-Ahles

Appendix D

Dear Dr. Chambers,

Thank you for your assistance with this manuscript. We would also like to express our thanks to all reviewers for taking the time to review the paper so many times. We have made the changes as requested (or as close as we can); the updates are described below and can be seen in the manuscript with Track Changes

Best,

Steve Politzer-Ahles

Reviewer 1

Although I am aware that this was not in the pre-registered analysis plan, it is curious that the two coding conditions are not statistically compared - such that it could be said that including stimuli with inconsistent place articulation results in significantly lower accuracy than consistent place articulation? Even if this is not explored statistically, it seems this could be given some mention in the discussion. This is important to guide future research design, and also because the authors suggest that lower McGurk effects in Proverbio et al's study might in part have been due to the type of stimuli included.

I'm actually not sure how to do this within one statistical model, so what I have done instead is do a kind of indirect hack at a conceptually related comparison: seeing if the "ambiguous" stimuli (the ones coded differently in our two analysis schemes) pattern with the incongruent or the congruent stimuli. The logic is explained in the new paragraph at the end of the results section. I think this will also provide the same kind of guidance the reviewer is looking for, as it clearly shows that it makes sense to treat the "ambiguous" stimuli as congruent rather than incongruent.

2. Page 12, line 34 "If anything, is musicians are better than non-musicians at processing speech in noise" another very relevant reference here is Coffey et al "Speech-in-noise perception in musicians: A review"

Added.

3. Page 13 line 4 "The present study obviously rules out those conclusions: if musicians are indeed subject to the McGurk effect, then McGurk studies do not provide any evidence for brain-level functional reorganization as a result of musical training." This seems quite a bold claim considering this study still reported

between groups differences (albeit in the opposite direction to Proverbio and colleagues!). Differences between groups surely suggests that the McGurk effects might show some functional differences – although the exact nature of these would need brain imaging studies to confirm the nature of plasticity (which the authors acknowledge this study does not rule out).

We have updated this final paragraph to clarify that we just mean conclusions about *that kind of* functional reorganization are ruled out. As the reviewer correctly notes, our results may well be consistent with some other kind of functional reorganization, just not the kind the original study described.

Reviewer 3

The authors provide two comparisons between musicians and non-musician levels of mcgurk. CI says no diff, GLMM says yes, likely b/c of odds-ratios are more sensitive to changes in probability scale. It would be helpful for a non-specialist reader to have this apparent (not actual) contradiction explained away.

Thank you for pointing this out; we have added the following text to clarify:
"(While the significant model coefficient for this effect may appear to contradict the confidence interval reported above, which includes 0%, the source of this apparent discrepancy is that the reported confidence interval is converted into percentage units, whereas the model coefficient and associated p-value is based on odds ratios, which are more sensitive to differences in probability. The calculation of these values is shown in the data analysis code on OSF.)"

Additionally, the authors should make clear which parameters are being referred to when presenting the coefficients from the GLMM. For instance, in the sentence, "The statistical analysis confirmed that musicians have a highly significant McGurk effect ($b=-2.34$, $z=-18.37$, $p<.001$), and that it in fact is significantly larger than the non-musicians' McGurk effect ($b=0.46$, $z=2.72$, $p=.007$)" I would think the first parameter is the main effect (intercept term) of condition (Incong < audio) and the second parameter is the interaction between condition and group, but this should be made explicit. Similar for secondary analysis, page 10, lines 38-42.

We have added the requested clarification. In the first part pointed out (the "Replication analysis" section) we now have the following text: *"The statistical*

analysis confirmed that musicians have a highly significant McGurk effect ($b=-2.34$, $z=-18.37$, $p<.001$), based on the simple effect coefficient for stimulus condition (which refers to the difference between audio-only and audiovisual incongruent for musicians, since musicians were coded as the baseline level of the group factor). Furthermore, musicians' McGurk effect was in fact significantly larger than the non-musicians' McGurk effect ($b=0.46$, $z=2.72$, $p=.007$), as shown by the condition-group interaction coefficient." In the second part (the "Targeted analysis" section) we have just added the names of the coefficients in the same parentheses that report the numerical coefficients and stuff.

The authors should remind the reader of the relative sample sizes in the first sentence, rather than just referring to the proportional increase.

Done! We now have: "In a pre-registered study with more than double 60 musicians and 60 non-musicians (compared to 30 musicians and 30 non-musicians in the original study)..."

I found the first paragraph of the discussion hard to follow as it tallies similarities and differences between the results. A figure here with appropriate confidence intervals would bring great clarity and provide a useful summary of the replication attempt.

This is a good idea and we are willing to do whatever is necessary to make the writing to follow. But we aren't sure how to make a graph like this that will clarify things. It would be easy for us to make a graph showing our effect sizes and CIs. But, as explained in the Discussion, we have little idea what the real effect sizes and CIs for Proverbio et al. are, so I'm not sure what we would put in this graph for comparison. Nevertheless, the reviewer's point that this paragraph is hard to follow is well taken, so we agree something should be done. We have attempted to satisfy this request by adding a schematic graph—not showing actual effect sizes and CIs (since those are not known for Proverbio et al.) but showing a sort of idealization of where we think those effects probably fall relative to ours (essentially it's just a visualization of the same patterns that are verbally described in that paragraph).

The second paragraph also devolves a bit into diatribe listing the problems with the original study's numerical analysis. I think this is a serious issue, and thus should not be relegated to a, quite long, 79-word parenthetical remark. Instead, the authors should create separate paragraphs for each reason a direct comparison of the numerical results is difficult/impossible.

We have divided this paragraph into several paragraphs now and walked through the issues in more detail.

A rehash of major differences between the original and replication study would be helpful, perhaps as a table. This allows readers to form their own opinion about how closely the replication followed the original. I think the point about musicians (in general) perceiving the McGurk illusion is well-demonstrated by this new paper, but perhaps there were things in the original study that made a null-finding more likely, e.g., sample/population characteristics, stimuli used etc. Again, having this information in a table will facilitate easy comparison and provide some structure for the discussion.

We have now added such a table in the Discussion section.

We have also done each of the minor suggestions.